# Exposure to Healthy Weight Information on Short-Form Video Applications to Acquire Healthy Weight-Control Behaviors: A Serial Mediation Model

**DOI:** 10.3390/ijerph20064975

**Published:** 2023-03-11

**Authors:** Donghwa Chung, Yanfang Meng

**Affiliations:** 1School of Journalism and Communication, Shanghai University, Shanghai 200444, China; 2Network and New Media, Beijing Institute of Graphic Communication, Beijing 102627, China

**Keywords:** healthy weight information, healthy weight-control behaviors, short-form video, third-person effect theory

## Abstract

This study explored the effects of Chinese college students’ (20–34 years old) exposure to healthy weight information on short-form video applications on their intention to acquire healthy weight-control behaviors (reducing high-fat diet intake, accessing physical activity to control body weight, etc.). Specifically, this study investigated the direct and mediated effect on such a relationship via healthy weight awareness, the first-person effect, and perceived herd. The data were collected using a web-based survey and thoroughly tested questionnaire with a sample of 380 Chinese college students. Hierarchical regression, parallel mediation, and serial mediation analysis were applied to test the hypotheses. The results indicated that healthy weight awareness, first-person effect, and perceived herd all played mediator roles that induced the relationship between Chinese college students’ exposure to healthy weight information and their intention to acquire healthy weight-control behaviors. In addition, healthy weight awareness and the first-person effect sequentially mediated this relationship.

## 1. Introduction

In November 2022, recurrent outbreaks of COVID-19 and increased confirmed daily new cases urged the Chinese Center for Disease Control (CDC) to issue a stay-at-home order to reduce the spread of COVID-19 [1]. The prolonged lockdown limited outdoor activities, which reinforced Chinese citizens’ lifestyle problems [2]. A number of studies stressed that the COVID-19 pandemic led to individuals adopting an unhealthy diet and poorer lifestyle; the remaining question was whether college students gained weight during this time [3,4]. Tavolacci et al. [5] found that college students’ overweight and obesity increased during the COVID-19 pandemic. Similarly, an investigation also demonstrated that almost 45.5% of university students experienced weight gain during the one month of lockdown in Malaysia. Such healthy-weight-maintenance-related issues caught the attention of the China Health National Health Commission. In response to this circumstance, the departments cooperated with social media platforms, encouraging content creators to promote healthy weight lifestyles and preventive behaviors to the media audiences [6,7]. However, it remains unclear how exposure to healthy weight information on cutting-edge new media platforms (e.g., short-form video applications) may impact Chinese media users’ acquisition of healthy weight-control behaviors during the ongoing COVID-19 pandemic in China.

Chinese short-form video applications have skyrocketed in popularity globally in recent years [8]. One of the latest investigations has indicated that short-form video applications have mostly been promoted to Chinese young media users, and are widely accepted by college students (20–34 years old) [9]. The applications are mainly recognized as entertainment platforms, where users create short-form videos or perceived content from other accounts [10]. Aside from their entertainment features, one of the most recent studies has indicated that short-form video applications offer great opportunities for disseminating health-related information [11]. Specifically, these applications’ users have demonstrated higher perceived credibility in regard to health-related short-form videos, which has further encouraged their intention to use these applications to obtain health information [12,13,14]. However, prior studies have failed to investigate such mechanisms in Chinese users’ acquisition of healthy weight-control behaviors. As discussed above, investigating such a phenomenon is critical for two reasons. First, Chinese health institutions are eager to know whether healthy weight promotion on short-form video applications is effective, and what the likelihood is of individuals being willing to acquire healthy weight-control behaviors [15]. Second, no empirical research has fully investigated the mechanisms that lead to Chinese short-form video users’ acquisition of healthy weight-control behaviors to date.

To fill this gap, the current study applied the third-person effect theory as a guide to further investigate the direct effect between Chinese college students’ (20–34 years old) exposure to healthy weight information (EHWI) on short-form video applications, and their intention to acquire healthy weight-control behaviors (IAHWCB) (H1). This study also identified possible mediating mechanisms in this relationship, based on health awareness, first-person effect (FPE), and perceived herd (PH) perspective (H2–H5).

## 2. Literature Review

### 2.1. TPE

TPE in communication arises when an individual exposed to a mass-media message perceives that message as being more impactful or persuasive to others than to him or herself [16]. In general, the TPE has been explored through two aspects in the field of communication and mass media. The first aspect is the so-called “perceptual hypothesis”: identifying a self–other discrepancy regarding exposure to persuasive media messages through mass media. In other words, the perceptual hypothesis explores whether people believe that the media content has the greatest influence on “them” rather than on “me”. The second aspect refers to the “behavioral hypothesis”. In most cases, individuals’ decisions are affected by their presumption of media impact (TPE) [17]. Therefore, this hypothesis refers to individuals’ presumption that the effect on behavioral intentions is greater for others compared to themselves.

Previous studies have explored TPE in the context of news about bird flu outbreaks [18], voters’ attitudes toward polls in the 2008 U.S. presidential election [19], the impact of fake news online on voter decisions [20], individuals’ perception of fake news during the COVID-19 outbreak [21], and the media’s influence on individuals’ vaccination against COVID-19 [22]. Although existing studies have examined both perceptual and behavioral hypotheses, there are certain gaps that remain unsolved. For instance, most of the studies overlooked how socially undesirable information increases TPE (e.g., pornography- and violence-related messages), whereas only a few studies have identified how socially desirable information decreases the self–other discrepancy [23]. Moreover, numerous scholars have broadly investigated TPE in Chinese media (such as Chinese TV, print media, social networking websites, and social media) [24,25,26]. However, there are still very few studies discussing the media effect of short-form video applications. Lastly, only a handful of health communication studies have explored the link between TPE and health behaviors, namely, vaccine hesitancy [27], COVID-19 prevention [28], and safer-sex consciousness [29]. However, the academic literature has paid little attention to how TPE leads individuals to acquire healthy weight-control behaviors.

Prior behavioral studies have argued that not all self–other perceptual discrepancies are considered TPE [23,30]. Scholars have identified an induced likelihood for individuals to presume that a media message is more impactful and persuasive to themselves compared to others [31]. Moreover, individuals show stronger effects for themselves than others when receiving a message that they presume to be socially desirable [23]. This effect is identified as reverse TPE, also called FPE.

Among the social media platforms in China, users have deemed TikTok (Douyin) as an application with large amounts of socially desirable content. This is one of the reasons that individuals continue to use TikTok, as well as for the induced social rewards and self-presentation [32]. For instance, a search of the keywords “Slimbody” and “Fitness inspiration” in December 2022 on Douyin demonstrated that the recommended videos (determined by the number of likes and comments) were viewed between 17 million and 20 million times. Most of the short-form video content could be categorized as the flat tummy challenge, athletic women showcasing workout routines, and weight-loss transformations before and after. Chinese short-form video users tend to assume that these types of videos are socially desirable content. This may increase users’ evaluation of the message’s impact on themselves compared to other users. According to Bavikatty’s [33] study, when female TikTok users are exposed to short-form video content with thin body types, they are likely to perceive the impact of the thin body ideal, and tend to compare their body type with that of influencers. However, little is known about whether reversed TPE exists in the context of healthy weight promotion on Chinese short-form video applications. Therefore, based on the concept of FPE, the current study explores the effect of perceived exposure to healthy weight information on short-form video applications on Chinese college students’ FPE (perceptual hypothesis). This study also explores how this will further reinforce them to acquire healthy weight-control behaviors (behavioral hypothesis).

### 2.2. Effect of EHWI on IAHWCB

Popular Chinese short-form video applications have a powerful algorithm feature that fully meets the needs of Chinese college students [8]. Moreover, a previous study has demonstrated that Chinese citizens’ use of Douyin (overseas version of TikTok) has greatly reinforced their interest in managing health-related content and disease-prevention knowledge [34]. Moreover, in an effort to spread healthy weight-control behavior, short-form video applications have collaborated with the Chinese health department to spread healthy weight information (e.g., weight-control and engagement in weight loss behaviors) [6]. In this study, healthy weight information (EHWI) refers to the frequency of individuals’ perceived coverage of healthy weight issues on short-form video applications.

Maintaining a weight within the normal range is critical to overall health. This is beneficial for preventing range of major diseases (e.g., heart disease, high blood pressure, type 2 diabetes, and breathing difficulties) [35]. Prior studies have indicated that the prolonged COVID-19 pandemic has worsened individuals’ weight-management maintenance. For instance, poor dietary habits [36], binge eating behaviors [37], and unwillingness to engage in physical activities have increased [38]. Investigating this phenomenon is critical to understand why individuals developed such behaviors during the COVID-19 pandemic. It is also urgent to know whether healthy weight campaigns in the media have successfully changed perceptions or behaviors. Lampard [39] categorizes healthy weight-control behaviors into six categories. Based on the individual frequency of activities dedicated to losing weight or preventing weight gain undertaken in the last year, the six specific behaviors are engaging in exercise, consuming more fruits and vegetables, controlling the intake of high-fat food, reducing the intake of sweets, avoiding drinking soda pop, and controlling portion sizes for each serving. In this study, the intention to acquire healthy weight-control behaviors (IAHWCB) refers to the possibility of an individual actively engaging in weight-control behaviors (reducing high-fat diet intake, increasing physical activity to control body weight, etc.).

A previous study has demonstrated that exposure to weight-stigmatizing media was positively associated with British female users’ weight loss intentions [40]. Similarly, individual exposures to weight-related messages have induced participation in weight-loss activities [41]. Moreover, a focus group study indicated that exposure to friends’ food-related posts on social media induced healthy diet behavior (healthy food choices) in young adults [42]. The following hypothesis is thus proposed.

**Hypothesis 1** **(H1).**
*EHWI has a significant and positive effect on Chinese college students’ IAHWCB.*


### 2.3. The Mediating Role of Healthy Weight Awareness

One of the key factors that predicts individuals’ engagement in health behaviors is health awareness [43]. In Mitchell and Prue’s [44] study, health awareness was conceptualized as general perception and knowledge of healthy behaviors. In the current study, healthy weight awareness (HWA) refers to Chinese college students’ healthy weight-related perception and knowledge. Previous studies have shown that mass-media exposure strongly predicted individuals’ maternal health awareness [45]. Furthermore, awareness of healthy food intake increases adults’ intentions to engage in weight management [46]. Similarly, a previous health communication study found that college students’ exposure to health news stories on TV is positively associated with their health consciousness (health awareness) [47], which further increased individuals’ healthy food-choice behavior [48]. In sum, it is logical that Chinese college students’ EHWI would increase their level of HWA. This would further affect IAHWCB. The following hypothesis is thus proposed.

**Hypothesis 2** **(H2).**
*HWA mediates the relationship between EHWI and Chinese college students’ IAHWCB.*


### 2.4. The Mediating Role of FPE

A handful of digital media and behavior-change studies explored the likelihood that media users evaluate a media message as being more impactful and persuasive to themselves compared to others [23,31]. Such an effect is identified as FPE. However, in this study, FPE refers to when individuals are EHWI and assume that healthy weight issues are much more important compared to other users. Previous studies have demonstrated that exposure to environmental videos increases individuals’ FPE [23]. Additionally, this impacts their intention to partake in COVID-19 preventive behavior [49]. Moreover, Day [50] found that exposure to public affairs advertisements amplified the individual degree of FPE. Furthermore, increased environmental risk (to self) induces the intention to promote pro-environmental news articles via social media [51].

**Hypothesis 3** **(H3).**
*FPE mediates the relationship between EHWI and Chinese college students’ IAHWCB.*


### 2.5. The Mediating Role of PH

Sundar et al. [52] indicated that selecting information, liking, and sharing features on social media increase individuals’ perceived control and sense of agency. Moreover, such perceived psychological effects have a positive impact on their knowledge, attitudes, and behavior [53]. One health communication study indicated that frequent social media engagement improved individuals’ HIV awareness and attitude [54]. Parallel to this investigation, numerous studies have also investigated how passive exposure to social media amplifies individuals’ sense of health. For instance, Waddell and Sundar [55] suggested that the bandwagon effect occurs when individuals are exposed to the apparent opinion of the crowd via online comments. Specifically, observing social media metrics (such as likes, comments, and shares) is positively associated with changes in social media users’ attitudes and behaviors [56]. Similarly, scholars have identified the concept of PH as a lens by which media users’ high-risk behaviors can be investigated [57,58]. PH refers to an individuals’ observation of a great number of others performing a certain behavior. This increases the chance that individuals will perform the same behavior they have seen previously [57]. Huang [58] conceptualized this term as WeChat users’ willingness to share a piece of information after being exposed to the most-shared pieces of information. This study investigates how PH manifests in the context of Chinese college students’ IAHWCB; in other words, whether the possibility of imitating others increases if the students witness highly rated (likes and shares) short-form videos of others engaging in healthy weight-related behaviors. Therefore, in this study, PH refers to Chinese college students’ willingness to engage in healthy weight-loss behaviors that are shared and liked by a large number of others on short-form video applications. Previously, communication researchers have demonstrated that exposure to the number of “likes” embedded in online comments triggers individuals to process and evaluate the comments [59]. Ultimately, this increases individuals’ intention to acquire preventive skin behaviors [60]. Moreover, Lim et al. [61] found that exposure to nonprofit organizations’ advertisements increased the bandwagon effect (PH), which increased individuals’ intention to engage in non-profit donation. Thus, it is logical that Chinese college students’ EHWI would increase their level of PH. This will further affect their IAHWCB. The following hypothesis is thus proposed.

**Hypothesis 4** **(H4).**
*PH mediates the relationship between EHWI and Chinese college students’ IAHWCB.*


### 2.6. The Serial Mediating Role of HWA and FPE

The above literature demonstrates the direct and mediated effects of EHWI on Chinese college students’ IAHWCB. However, the previous literature suggested a serial mediation mechanism in this relationship. Debatin et al. [62] indicated that Facebook users were more likely to participate in protective behaviors if they believed that they could experience negative consequences after using the social media platform. Similarly, the elders’ perception of COVID-19 reduced their degree of optimistic bias regarding COVID-19 infection [63]. In other words, individuals’ perceived awareness increased their presumption that they have a high risk of contracting COVID-19 compared to others (FPE). Considering the studies discussed above, this study aims to further examine the sequential mediation chain of HWA and FPE in the relationship between EHWI and IAHWCB. Hence, the following hypothesis is proposed (Figure 1).

**Hypothesis 5** **(H5).**
*The relationship between EHWI and Chinese college students’ IAHWCB is sequentially mediated by HWA and FPE.*


## 3. Materials and Methods

### 3.1. Questionnaire Design

The current study adopted 5 measurements (e.g., EHWI, HWA, FPE, PH, and IAHWCB) from previous studies. Nonetheless, the measurements have not yet been applied in the context of Chinese culture. To obtain higher precision and accuracy in the measurements, the current study carefully made modifications based on the following rules. First, each measurement was translated from the original English form into Chinese by two language experts. Once the translation was completed, all researchers compared the original to the translated measurements several times, and came to an agreement regarding the translation’s precision and accuracy [8]. Second, face validity methods were applied to adjust and remove inappropriate items [64]. Therefore, three experts in the field of journalism and media were invited and asked to participate in this evaluation [8]. Lastly, a pre-test was arranged with 10 volunteers, who were recruited from both Shanghai University and the Beijing Institute of Graphic Communication. All volunteers were requested to provide recommendations for further revision, for example, to state whether any items were unreadable or ambiguous [65]. We then uploaded the finalized survey on the online survey platform “Wenjuanxing”. This platform provides a sampling pool of nearly 260 million registered users in China (excluding Tibet and Qinghai province in China). The platform provides great data quality and options that best fit the researcher’s needs. This has been widely adopted in academic research related to China [66,67]. Permission to conduct the current study was reviewed and approved by the Beijing Institute of Graphic Communication Academic Committee (20221129). The statement of permission was included at the beginning of the online questionnaire. The data were collected from 1 October to 1 December 2022 via Wenjuanxing. Among the 450 potential respondents who received the survey link, 380 filled out the questionnaire.

### 3.2. Measurement of Variables

All the measurements were evaluated using a five-point Likert scale. Answers ranged from strongly disagree to strongly agree or from very rarely to very frequently. EHWI was adopted according to the definition of Shen et al. [68]. This was measured as an index of how often respondents watch the topics “fat and sugar reduction,” “Low-calorie recipes,” “Nutritionist recommendation for weight loss,” and “fat-burning workout,” in short-form videos on the following six applications: (a) Douyin, (b) WeChat short video, (c) Kuaishou, (d) Watermelon video, (e) Volcano video, and (f) Meipai. Those items were added to create an EHWI measurement on a five-point Likert scale (M = 2.80, SD = 1.00, α = 0.86).

To observe HWA, the current study adapted the health-awareness scale of [44]. Four items were assessed by five-point Likert scale, with statements such as “Failure to control weight can lead to a number of diseases (e.g., cardiovascular disease, digestive system function),”and “Obesity causes irreversible diseases and increases the risk of getting cancers.” (M = 4.00, SD = 0.90, α = 0.90).

FPE measures how the importance of maintaining a healthy weight could affect participants, with four items derived from [23]. Participants were asked to respond to a five-point Likert scale. Items included statements such as “I understand that maintaining healthy weight is critically important,” and “I am aware of the consequences of not controlling my weight could lead seriously damage to my body” (M = 3.50, SD = 1.10, α = 0.90).

PH was measured as an index of individuals’ willingness to engage in healthy weight-loss behaviors, which are shared and liked by many others on short-form video applications. The four items were revised from the scale of PH from a previous study [58]. Examples of these statements are “The more “likes” a short-form video gets, the more I will be willing to control my weight and stay healthy,” and “The more “shares” a short-form video gets, the more I will be willing to control my weight and stay healthy” (M = 3.20, SD = 1.12, α = 0.93).

IAHWCB was operationalized with four items, which were adapted from Lampard [39]. Example items are “I intend to engage in aerobic or anaerobic exercise in the near future,” and “I have it in my mind that I would limit the consumption of high sugar foods and beverages” (M = 3.11, SD = 0.90, α = 0.80).

## 4. Results

### 4.1. Descriptive Data

Three-hundred-and-eighty valid responses were collected. The demographic characteristics of the survey participants are outlined in Table 1. Respondents were mostly Female (N = 197, 51.8%), single (N = 347, 91.3%), and were either undergraduates (N = 246, 64.7%) or master’s students (N = 59, 15.5%). The respondents’ age range varied from 18 to 21 years old (N = 304, 80.0%), and their monthly income ranged from RMB 1000 to RMB 6999 (N = 219, 57.6%). The bivariate association among the independent, dependent, and control variables (gender, education, and income) can be seen Table 2.

### 4.2. Direct and Mediated Effect Test

To test Hypothesis 1, this study applied a hierarchical regression analysis with IAHWCB as a dependent variable. Gender, education, and income were entered into the first block as controlling confounders, and EHWI was entered in the second block. The effect of Chinese college students’ EHWI on IAHWCB was significant (β = 0.45, *p* <0.001). Figure 2 indicates the standardized coefficients and significance for each path in the hypothesized model.

Hayes’s PROCESS macro (Model 4) was applied to test the mediation analysis of HWA, FPE, and PH on the relationship between EHWI and Chinese college students’ IAHWCB. This study then applied bootstrapping to obtain bias-corrected 95% confidence intervals to make statistical inferences about specific indirect effects [69]. In the first mediation model, HWA positively predicted IAHWCB (β = 0.10, *p* < 0.01). In addition, EHWI positively predicted IAHWCB (β = 0.43, *p* < 0.001). The mediation effect test showed an indirect effect of EHWI on IAHWCB, mediated by HWA (β = 0.02, *p* < 0.001, 95% CI [0.01, 0.05]). Therefore, the indirect effect was significant, and the partial mediation effect of HWA was confirmed. The second mediation model’s result indicated that FPE positively predicted IAHWCB (β = 0.32, *p* < 0.001). Meanwhile, EHWI positively predicted IAHWCB (β = 0.27, *p* < 0.001). The mediation effect test showed an indirect effect of EHWI on IAHWCB, mediated by FPE (β = 0.19, *p* < 0.001, 95% CI [0.13, 0.25]). Therefore, the indirect effect was significant and the partial mediation effect of FPE was confirmed. The third mediation model demonstrated that PH positively predicted IAHWCB (β = 0.29, *p* < 0.001). Furthermore, EHWI positively predicted IAHWCB (β = 0.30, *p* < 0.001). The mediation effect test showed an indirect effect of EHWI on IAHWCB, mediated by PH (β = 0.16, *p* < 0.001, 95% CI [0.10, 0.22]). Thus, the indirect effect was significant, and the partial mediation effect of PH was confirmed.

### 4.3. Serial Mediating Effects Test

The PROCESS macro (Model 6) was applied to test the serial mediating effects of HWA and FPE in the relationship between EHWI and Chinese college students’ IAHWCB. The results demonstrated that EHWI positively predicted HWA (β = 0.24, *p* < 0.001), FPE (β = 0.49, *p* < 0.001) and IAHWCB (β = 0.27, *p* < 0.001) (Table 3). HWA positively predicted FPE (β = 0.43, *p* < 0.001), but did not predict IAHWCB (β = 0.04, *p* = 0.40). Lastly, FPE positively predicted IAHWCB (β = 0.34, *p* < 0.001). A total of 5000 bootstrap estimates were applied to construct the 95% confidence intervals for the indirect effects [70]. The serial mediating effect of HWA and FPE on the relationship between EHWI and IAHWCB was significant (β = 0.03, *p* < 0.001, 95% CI [0.02, 0.06]).

## 5. Discussion

By applying the third-person effect theory as a guideline, this study further investigated the effect of EHWI on short-form video applications on IAHWCB among Chinese college students (20–34 years old). The main objective of the current study is an in-depth exploration of the direct and indirect effects of critical factors (HWA, FPE and PH) on such a relationship. Moreover, this study aims to further examine the sequential mediation chain of HWA and FPE in the relationship between EHWI and IAHWCB.

In terms of direct effects, EHWI was positively associated with Chinese college students’ IAHWCB. The finding is in line with previous studies that indicated that exposure to media information amplified individuals’ healthy weight-related behaviors [40,41,42]. For instance, Pan [41] suggested that the more individuals are exposed to weight-related messages, the higher their willingness is to participate in weight-loss activities. Regarding mediated effects, the finding of H2 indicated that HWA mediated the relationship between EHWI and Chinese college students’ IAHWCB. This finding is consistent with prior studies [45,47,48], which found that college students’ exposure to health-related news content on TV increased their degree of health awareness. Ultimately, this increased their adoption of healthy food-choice behaviors [47,48]. FPE explained that individuals are more likely to presume that a media message is more persuasive to themselves compared to others [31]. A prior study has indicated that exposure to public affairs-related information induced a higher level of FPE in individuals. Ultimately, increased FPE amplifies people’s intention to share pro-environmental-related news on social media [51]. It is logical that FPE could play the role of the mediator in the relationship between perceived information and individuals’ intention to acquire behaviors. This study also found that FPE mediated the relationship between EHWI and Chinese college students’ IAHWCB (H3). This finding is consistent with the previous literature, suggesting that individuals’ exposure to environmental-related videos increased their degree of FPE [23]. Moreover, Mesch et al. [49] found that FPE is positively associated with the intention to engage in COVID-19-preventive behavior. Lastly, the findings of H4 demonstrated that PH mediated the relationship between EHWI and Chinese college students’ IAHWCB. This finding is in line with earlier studies, which have shown that individuals who are exposed to highly liked online comments have a more positive judgment of the comments [59]. Additionally, individuals’ PH encouraged them to acquire preventive skin behaviors [60].

Individuals’ maintenance of weight-loss behavior is considered to be a complex interaction that involves multiple mechanisms. Specifically, cognitive factors are recognized as unique factors that influence such behaviors [71]. For instance, when individuals perceive a health threat, their cognitive processes tend to deal with the threat by themselves, as compared to other cognitive processes. Additionally, a recent study found that individuals who were exposed to fear-related messages had an increased health awareness of COVID-19, which amplified their degree of vigilance [72]. Vigilance is defined as being very careful to notice things or watchful for whatever may occur [73]. This cognitive factor is seemingly related to FPE, in which individuals tend to presume the degree to which healthy weight issues are much more important than others. Ultimately, this effect was positively associated with the intention to promote pro-environmental news articles via social media [51]. In sum, as the findings of the current study demonstrate, Chinese college students’ EHWI on short-form video applications increases their level of HWA, which reinforces their degree of FPE. Furthermore, increased FPE is correlated with IAHWCB.

The recommended practical implications are as follows. Firstly, EHWI can drive HWA, which further reinforced Chinese college students’ IAHWCB. Therefore, this study suggests that Chinese short-form video applications motivate content creators to create various types of healthy weight-related short-form videos. This may further impact Chinese college students’ salience of healthy weight literacy and related knowledge. Secondly, the prior study suggested that social networking provided peer social support, which encouraged others to acquire weight-loss behaviors [74]. Moreover, Chung et al. [75] indicated that the fact that college students frequently engage in conversation via Facebook led to their intention to become involved in weight-loss behaviors. Thus, Chinese university teachers and family members should encourage college students to connect and actively engage with their peers to include healthy weight-related topics in short-form video applications (e.g., commenting and sharing educational content together). Lastly, a previous study demonstrated that during the COVID-19 pandemic, many misleading short-form videos were disseminated [76]. In most cases, these platforms contain both verified and unverified health information. In response to this issue, a previous internet and educational study has suggested improving college students’ information literacy, so that they have the ability to determine what is correct information, and what is misleading or fake information [77]. Therefore, Chinese college teachers should be encouraged to utilize their courses to discuss fake information on short-form video applications and how to recognize it.

There are several limitations to this study that should also be mentioned. First, the current study was a cross-sectional study. This is limited to the identification of each factor’s causal relationship (both direct and mediated effects). Future studies should conduct deeper investigations, such as longitudinal quantitative studies. Second, the current study IAHWCB was adopted from Lampard [39], and there are only four items representing Chinese college students’ intention to acquire healthy weight-control behaviors. However, the limited number of items makes it difficult to measure their actual intention to engage in all types of healthy weight-related behaviors. Thus, more precise measurements should be operationalized in future studies. Despite this study expanding the knowledge of information avoidance behavior by empirically examining the effect of three key determinants on the relationship between EHWI and IAHWCB, it has not yet fully considered health as a “dialogic process” which triggers individuals’ pathologies or actions in health communication [78]. Therefore, it is important to further explore how individuals are persuaded by health information via linguistic features. For example, one unique study applied Proppy’s architecture and fully evaluated how political information features (such as N-gram, Lexicon, and vocabulary richness) could gain individuals’ attention and further change their opinions [79]. Therefore, examining the effect of persuasion of EHWI on individuals’ IAHWCB will extend the current knowledge of individuals’ IAHWCB. In addition, future research needs to further investigate the influence of normative and cognitive factors: for instance, how injunctive norms and descriptive norms directly, indirectly or sequentially mediate the relationship between EHWI and Chinese college students’ IAHWCB. Additionally, they should test whether an individual’s optimistic bias (optimistic bias is people’s tendency to overestimate their likelihood of experiencing positive events and underestimate their likelihood of experiencing negative events in the future) is positively or negatively associated with IAHWCB. These potential factors will fill the gap in understanding the effect of short-form video media on media users’ behavior changes.

## 6. Conclusions

The current study took a critical step to explore direct and mediated mechanisms in the relationship between EHWI on short-form video applications IAHWCB among Chinese college students (20–34 years old), a critical and understudied phenomenon that has threatened this population’s motivation to maintain a healthy weight amid COVID-19 lockdowns in China. First, the current study confirms that EHWI has a direct impact on Chinese college students’ IAHWCB. Second, this study also found that a mediated effect exists on the relationship between EHWI and IAHWCB via HWA, FPE, and PH. That is to say, unlike other determinants, EHWI on short-form applications increases Chinese college students’ HWA, FPE, and PH. Ultimately, these factors amplify their IAHWCB. Lastly, the result also demonstrated that there is a serial mediation mechanism in HWA and FPE for linking EHWI and Chinese college students’ IAHWCB, which successfully supports the previous evidence [62,63].

## Figures and Tables

**Figure 1 ijerph-20-04975-f001:**
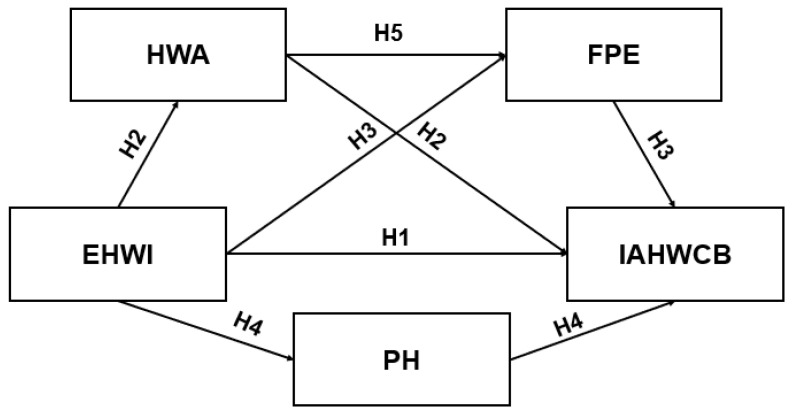
A model of predictors of intention to acquire healthy weight-control behaviors.

**Figure 2 ijerph-20-04975-f002:**
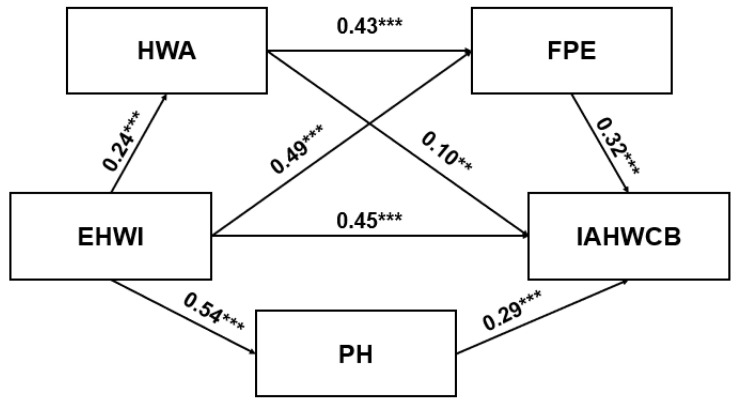
The effects of predictors of intention to acquire healthy weight-control behaviors. ** *p* < 0.01 *** *p* < 0.001.

**Table 1 ijerph-20-04975-t001:** Key demographic characteristics of the survey participants.

Demographics	Item	Frequency	Percent
Gender	Male	183	48.2%
	Female	197	51.8%
Education Level	High school	21	5.5%
	Pre-college	43	11.3%
	Undergraduate	246	64.7%
	Master’s	59	15.5%
	Ph.D.	11	2.9%
Age	18–21 years old	304	80.0%
	22–26 years old	76	20.0%
Marital status	Single	347	91.3%
	Married	33	8.7%
Monthly Income	1000–6999 RMB	219	57.6%
	7000–14,000 RMB	90	23.6%
	14,000–49,999 RMB	57	15.0%
	>50,000 RMB	14	3.7%
Total		380	100%

**Table 2 ijerph-20-04975-t002:** Correlations matrix of key variables.

Variables	1	2	3	4	5	6	7	8
EHWI	–							
HWA	0.25 **	–						
FPE	0.58 **	0.51 **	–					
PH	0.47 **	0.36 **	0.62 **	–				
IAHWCB	0.47 **	0.30 **	0.53 **	0.48 **	–			
Gender	−0.98	0.13 *	−0.10 *	−0.70	−0.15 *	-		
Education	−0.10	0.61	−0.03	−0.07	0.25	0.22 **		
Income	−0.94	0.24	−0.64	−0.04	0.01	0.74	0.23 **	-

Note. EHWI = exposure to healthy weight information, HWA = healthy weight awareness, FPE = first-person effect, PH = perceived herd, IAHWCB = intention to acquire healthy weight-control behaviors. * *p* < 0.05 ** *p* < 0.01.

**Table 3 ijerph-20-04975-t003:** Summary of mediating effect tests.

Indirect Effects	B	*p*	LLCI	ULCI
Total: EHWI → IAHWCB	0.19	*p* < 0.001	0.13	0.25
EHWI → HWA → IAHWCB	−0.01	*p* > 0.05	−0.04	0.01
EHWI → FPE → IAHWCB	0.17	*p* < 0.001	0.11	0.23
EHWI → HWA → FPE → IAHWCB	0.35	*p* < 0.001	0.02	0.06

## Data Availability

Not applicable.

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
