# Peer review of "Exposure to Healthy Weight Information on Short-Form Video Applications to Acquire Healthy Weight-Control Behaviors: A Serial Mediation Model"

_ijerph, 2023, doi:10.3390/ijerph20064975_

Round 1

Reviewer 1 Report

The article is very interesting and useful for several field of investigation. In particular the field of psychology about human health studies (see the suggestion below about the concept of health). In fact I suggest to the authors to include in their discussion and limits of the research (or future studies), the analysis of Da San Martino Giovanni and their colleagues to expand the study of the effect of persuasion.

Barron-Cedeno A., Jaradat I., Da San Martino G. and Nakov P. (2019). Proppy: Organizing the news based on their propagandistic content. Information Processing and Management, 1849-1864.

Turchi, G. P., Orrù, L., Iudici, A., & Pinto, E. (2022). A contribution towards health, Journal of Evaluation in Clinical Practice, 28, 717-720. https://doi.org/10.1111/JEP.13732

Reviewer 2 Report

The study on Exposure to Healthy Weight Information on Short-Form Video Applications to Acquire Healthy Weight-Control Behaviors is very important to the scientists and researchers reading this journal.

To improve the quality of the manuscript, the authors should clarify:

1. The relationship between body weight and Covid-19 is not clear, there are no supporting data. One of the reasons is that the 6 references do not have a date to verify their currency.

2. In this same sense, the conclusions are not related to covid-19.

3. The serial mediation model is very well structured. But even so, there is no correlation with other variables such as Age (Chinese university students (20 to 34 years old)), educational level, with covid-19, etc. This must be clarified, because it was not studied.

Six first references without format and date.

They are very old references: 18 of 2002, 35 of 1993 and 59 of 1997

Reviewer 3 Report

Congratulations, this is an interesting article with a valuable purpose for young people's health, and the results can contribute to the development of health promotion. However, I believe that to reinforce the results and improve the quality of the manuscript, the authors should pay attention to the following points:

  • Organize the first 6 references with date and appropriate format;
  • Update references with more than 20 years;
  • And, since they made reference to the probability of university students developing binge eating behaviors due to COVID-19, it would be important to clarify the more direct relationship of COVID-19 with the weight of students;
  • Conclusion: Rewrite, because it just repeats what was the purpose of the study and reinforces the idea that there is a gap in studies on the subject. I believe it is important to refer to the conclusion the authors reached about the effects of exposing college students to information. For example: what is the direct effect on healthy weight control behavior? What significant changes were produced in the behaviors?

Round 2

Reviewer 2 Report

It is especially important to recognize that the authors made all changes and corrections in your manuscript.

Accept in this format

Author Response

1.We agree with the reviewer on this important point. Therefore, We have defined the acronym in the text such as PH (see Line 91 to 93). 

2.Additionally, we have checked all acronyms along the text. We have summarized literature review. Thank you for pointing this out. 

3.We agree with the reviewer on this important point , Thus the points H1,2,3,4,5 have been inserted at the end of the introduction as objectives of the study. 

4.The sentences (such as “H2, H3 and H4 are fully supported”) have been moved in the discussion as well as any other comment on the results.